# Angle-Dependent Raman Scattering Studies on Anisotropic Properties of Crystalline Hexagonal 4H-SiC

**DOI:** 10.3390/ma15248751

**Published:** 2022-12-08

**Authors:** Zhe Chuan Feng, Dishu Zhao, Lingyu Wan, Weijie Lu, Jeffrey Yiin, Benjamin Klein, Ian T. Ferguson

**Affiliations:** 1Southern Polytechnic College of Engineering and Engineering Technology, Kennesaw State University, Marietta, GA 30060, USA; 2Center on Nano-Energy Research, Laboratory of Optoelectronic Materials & Detection Technology, Guangxi Key Laboratory for the Relativistic Astrophysics, School of Physical Science & Technology, Guangxi University, Nanning 530004, China; 3Hexagonal Scientific Lab, LLC, Dayton, OH 45459, USA

**Keywords:** silicon carbide, 4H-SiC, raman scattering, anisotropic property, angular dependence, raman tensor, selection rule

## Abstract

Raman scattering spectroscopy (RSS) has the merits of non-destructiveness, fast analysis, and identification of SiC polytype materials. By way of angle-dependent Raman scattering (ADRS), the isotropic characteristics are confirmed for c-face 4H-SiC, while the anisotropic properties of a-face 4H-SiC are revealed and studied in detail via combined experiments and theoretical calculation. The variation functional relationship of the angle between the incident laser polarization direction and the parallel (perpendicular) polarization direction was well established. The selection rules of wurtzite 4H-SiC are deduced, and the intensity variations of the A_1_, E_2,_ and E_1_ Raman phonon modes dependent on the incident angle are calculated, and well-matched with experimental data. Raman tensor elements of various modes are determined.

## 1. Introduction

Silicon carbide (SiC) has been recognized as an important material for a wide variety of high-power and high-temperature electronic applications [1,2,3,4,5,6,7]. SiC exhibits a large number (>250) of polytypes with different structural and physical properties, as well as possesses a wide range of applications [8,9,10,11,12,13,14,15]. The SiC polytypes have the same chemical composition but exhibit different crystallographic structures and stacking sequences along the principal crystal axis. Several important polytypes of SiC, such as 4H and 6H have C_6v_ crystallographic symmetry. In the “a” direction 4H– and 6H–SiC are almost identical (<1%) changes; however, the 4H polytype consists of four units in the c direction and the 6H consists of six units. Different polytypes have different band gaps, electron mobility, and other physical properties. 4H–SiC has attracted significant attention due to its high electron mobility and excellent thermal properties [16,17,18,19,20,21,22,23,24,25,26,27,28,29,30,31,32,33,34,35]. Especially in recent years (2020–2022), research activities on 4H-SiC have been extremely active and productive [6,11,12,13,14,15,24,25,26,27,28,29,30,31,32,33,34,35].

Since the late 1980s, high-quality and large-size wafers of both 6H– and 4H–SiC have come into industrial production. Gradually, the wafer size increased, up to 8 inches recently, and the wafer crystalline quality improved over time and is very widely used for device applications. Wafers of SiC are also promising substrates for nitride semiconductor growth due to their compatible lattice structure and similar thermal expansion coefficients, as well as for other materials.

Raman scattering spectroscopy (RSS) has the merits of non-destructiveness, fast analysis, and being free from special preparation for samples. It has been proven to be a useful and informative tool for the investigation and identification of SiC polytype materials. Furthermore, the Raman efficiency of SiC is high because of its strong covalence of the bonding, so the Raman signals can be easily detected. The Raman parameters, such as intensity, width, and peak frequency, provide plenty of information on the crystal quality. Up to now, Raman spectroscopy has been widely applied in the SiC bulk and epitaxial materials, especially for 4H-SiC [16,17,18,19,20,21,22,23,24,25,26,27,28,29,30,31,32,33,34,35].

Owing to the different vibrating types, the phonon intensity induced by the electric vector will vary from the polarization of the incident laser beam and will cause the multi-fold scattered light. The polarizability tensor of the wurtzite crystal [36,37,38,39,40,41] has been used to verify the Raman selection rule; however, the nonvanishing components of the susceptibility tensor should include the effects from different vibrating components—it is needed to further identify and quantify the anisotropic distribution of Raman signals.

Raman scattering spectroscopy (RSS) is an effective nondestructive method to characterize materials, to analyze the crystal structure, electronic band structure, phonon energy dispersion, and electron–phonon coupling [36,37]. By analyzing Raman spectra of wide bandgap semiconductor (WGS) materials, such as SiC [16,17,18,19,20,21,22,23,24,25,26,27,28,29,30,31,32,33,34,35], GaN [36], and AlN [37,38], one can gain knowledge of their crystal structure, carrier concentration, internal strain/stress and distribution, and other characteristic properties. It is interesting to see, during 2020-22, Raman scattering on 4H-SiC has contributed to many important subjects, such as basal plane stacking faults and associated partial dislocations [24], effects of ion-implantation [25,26,27,29], ionizing irradiation-induced atomic movement toward recrystallization [26], phase changes from excimer laser doping [32], mechanical stress and nitrogen doping on the defect distribution [31], and so on.

Angle-dependent or rotation Raman scattering has been applied to investigate WGS epitaxial GaN [36] and bulk AlN crystals [37,38,39,40,41]. From these experiments, the Raman tensors are derived, containing various information for the vibration symmetries, polarities, and anisotropies of crystals [39,40,41]. The so-called “phase difference” is commonly introduced as a phenomenological parameter and recognized as an intrinsic property of crystals, although its physics still remains for further investigation [39,40,41]. Recently, a combined experiment–theory study on the angle-resolved intensity of polarized micro-Raman spectroscopy by considering the birefringence effect was presented for the c-face of a 4H-SiC wafer, with planar plots for the relationship of Raman normalized intensity versus rotation angle [30]. As pointed out, so far, most of the Raman analysis is not based on the anisotropy of SiC, except for [30]. In early 1995, H. Horima et al. [16] reported Raman scattering from anisotropic LO phonon–plasmon-coupled (LOPC) mode in n-type 4H– and 6H–SiC.

In this paper, we employed the angle-dependent Raman scattering (ADRS) to study the phonon anisotropy property of the wurtzite 4H-SiC crystal both theoretically and experimentally. The angle-dependent polarized RS measurements were performed on the “a” face 4H-SiC crystal, and comparative c-face 4H-SiC, by adjusting the polarized vector of the incident and scattered laser light. Corresponding Raman selection rules are derived according to measured scattering geometries to illustrate the angle dependence. The angle-dependent intensities of phonon modes are calculated and compared to the experimental scattering intensities, yielding the Raman tensor elements of A_1_, E_1,_ and E_2_ phonon modes. These detailed theoretical calculation results on the Raman selection role, Raman tensor elements, and the variations of Raman spectral intensities of wurtzite 4H-SiC are in good coincidence with experimental data. To our knowledge, ADRS has been applied to GaN [36], AlN [37,38,39,40,41], MoS_2,_ and WS_2_ [39,40,41], and Raman tensors for these wurtzite wide gap materials are obtained also. Our current studies of ADRS and Raman tensors for 4H-SiC are intended to advance the knowledge on 4H-SiC and to promote the research and development for SiC more polytype materials and applications.

## 2. Experimental

The experimental 4H-SiC samples used in this work were purchased from CREE Company. A piece of un-doped a-face (1120) 4H-SiC with the size of 5 × 5 mm, was cut from a 2-inch wafer. Some c-face 4H-SiC experimental samples are pieces at the size of 5 × 5 mm or 10 × 10 mm, cut of commerce wafer materials (typically 1 3/8 to 2 inch).

Raman spectra were measured in backscattering geometry at room temperature (RT), by a Raman micro-spectrometer, connected with a Charge Coupled Device (CCD) for signal detection, named ‘Finder One’ from Zolix Company. It is equipped with four diode lasers possessing excitation wavelengths of 457, 532, 650, and 785 nm. In the current work, the 532 nm excitation laser was used, with incident light through the x50 object (with 0.55 numerical aperture and 4 mm focal length) focusing on the “a” surface or c-face of the 4H-SiC crystal sample, while the scattered light was collected via this same object. To make the scattered light parallel with or perpendicular to the incident light, a rotatable polarizer was inserted within the scattered light path. The sample stage could be rotated with 360°, i.e., the incident light could be rotated with 360° to obtain the angular dependent Raman spectra.

Figure 1 shows (a) the hexagonal wurtzite crystal structure and (b) the experimental arrangement. The XYZ coordinates with the crystalline axis of the 4H-SiC crystal. The *Z*-axis is along the c-axis of the wurtzite 4H-SiC crystal, i.e., (0001) direction. The *Y*-axis is along the normal direction of the a-face of 4H-SiC. The rotation angle θ is the angle of inclusion between the **ξ** axis and the *X*-axis. The incident light travels along the *Y*-axis, the SiC sample is rotated around the *Y*-axis, and rotation angle θ lies between the incident light polarizing vector in the X-Z plane and *X*-axis.

In this paper, we present the RT backscattering Raman spectra of the semi-polar a-face (1120) and c-face (0001) 4H-SiC crystal for the relationship between the lattice phonon vibration modes and the rotation angles of the wurtzite structure, which shows the crystal anisotropy or isotropy characteristics.

## 3. Results and Discussion

### 3.1. Angular-Dependent Raman Spectra from c-face 4H-SiC

Figure 2 presents the angle-dependent Raman spectra of the c-face (0001) 4H-SiC, measured at RT with the excitation of 532 nm. The hexagonal wurtzite 4H-SiC is a tetrahedrally coordinated semiconductor compound and belongs to the space group C^4^_6v_(P6_3_mc) in the primitive cell. All the atoms occupy sites of symmetry C_3v_ [7,11]. Main Raman active modes include E_2_ (LA), 203.5 cm^−1^; A_1_ (LA), 610.5 cm^−1^; E_2_ (TO), 777.0 cm^−1^; E_1_ (TO), 788.1 cm^−1^; and A_1_ (LO) 967.0 cm^−1^, as shown in Figure 2, in which all above Raman phonon modes are identical at different rotation angles of 0°–330° (0°, i.e., 360°). This indicates the isotropic characteristics of the c-face 4H-SiC crystal [30].

### 3.2. Angular-Dependent Raman Spectra from a-Face 4H-SiC

Figure 3 shows the rotation Raman spectra at the a-plane (1120) of 4H-SiC, in the wavenumber range of 580–820 cm^−1^ and with the angle rotated from 0° to 360° (step size of 5°), under (a) parallel and (b) perpendicular polarization, respectively. Within Figure 3, the A_1_ (LA), E_2_ (TO), and E_1_ (TO) modes are observed at 610.03 cm^−1^, 776.49 cm^−1,^ and 788.08 cm^−1^, respectively. The intensity and phase variations of these Raman modes and their dependences on the rotation angle are clearly shown. Further, there exists a shoulder peak in the close-right of E_2_ (TO) at ~783 cm^−1^, which is assigned as the A_1_ (TO) mode of 4H-SiC [29]. In addition, we provide a single point scan Raman spectrum of the a-face 4H-SiC, for reference, included in the Appendix A, with an inset and showing E_1_ (TA) at 266 cm^−1^, E_1_ (TA) at 610 cm^−1^, E_2_ (TO) at 775 cm^−1^, E_1_ (TO) at 788 cm^−1^, A_1_ (LO) at 976 cm^−1^ and LO-plasma coupling (LOPC) broadband in the right side of A_1_(LO) [17,18,19].

### 3.3. Raman Selection Rules

The 4H-SiC crystal is a compound semiconductor with a hexagonal wurtzite structure and belongs to the space group C^4^_6v_ (P63mc) with eight atoms in the primitive cell [7,11]. In its Brillion zone, the lattice vibration produces nine optic branches and three acoustic branches, including the A_1_, B_1_, E_1,_ and E_2_ vibration modes [7,11,17]. According to the group theory, A_1_ and E_1_ acoustic modes are Raman and infrared (IR) active. The A_1_ phonon is polarized along the *Y*-axis direction, while the E_1_ phonon is polarized within the XZ plane (see the Coordinate System in Figure 1). The E_2_ mode is only Raman active, while B_1_ belongs to the non-active mode. Due to the creation of polar bonds, there appear frequency shifts for A_1_ and E_1_ Raman modes with symmetries. The A_1_ and E_1_ Raman modes are split into the longitudinal phonon and transverse phonon modes, forming the A_1_ (TO, LO) and E_1_ (TO, LO) modes.

The Raman scattered light intensities can be expressed as [36,37,38,39,40,41]:(1)$$I\sim {\left|e_{s}\cdot R\cdot e_{i}\right|}^{2}$$
where e_i_ and e_s_ present the polarization vector for the incident and scattered light, respectively. R is the 2nd-order Raman tensor with the form of a 3 × 3 vector matrix [36,37,38,39,40,41]. This vector matrix represents the characteristics of Raman scattering phonon modes. The A_1_, E_1,_ and E_2_ Raman active modes in wurtzite structural materials are expressed below [36,37,38,39,40,41]:(2)$$R\left[A_{1}\right]=\left[\begin{array}{ccc} a & 0 & 0\\ 0 & a & 0\\ 0 & 0 & b \end{array}\right],$$
(3)$$R\left[E_{1}\right]=\left[\begin{array}{ccc} 0 & 0 & -c\\ 0 & 0 & c\\ -c & c & 0 \end{array}\right],$$
(4)$$R\left[E_{2}\right]=\left[\begin{array}{ccc} d & d & 0\\ d & -d & 0\\ 0 & 0 & 0 \end{array}\right],$$
where a, b, c, and d represent Raman tensor elements. In Raman backscattering measurements on the a-plane of 4H-SiC single crystalline material, referring to the hexagonal axis structure, the polarized vectors for the incident and scattered light can be expressed as:(5)$$e_{i}=\left(\begin{array}{c} \sin\left(\theta \right)\\ 0\\ \cos\left(\theta \right) \end{array}\right),$$
and
(6)$$e_{s}=\left(\begin{array}{c} \sin\left(\theta_{s}\right)\\ 0\\ \cos\left(\theta_{s}\right) \end{array}\right),$$
where θ (θ_s_) represents the angle between the sample *Y*-axis and the incident (scattered) light vector, respectively.

Here, we discuss the case of the parallel and perpendicular configuration, from which, according to the calculation based upon Equation (1), the Raman intensity at the surface of the a-plane from the 4H-SiC crystal can be expressed as:(7)$$I_{⫽}{(A}_{1})∼{\left|a\right|}^{2}\sin^{4}\theta +{\left|b\right|}^{2}\cos^{4}\theta +\frac{\left|a\right|\left|b\right|}{2}\sin^{2}2\theta \cos\left(\varphi_{a-b}\right),$$
(8)$$I_{⊥}\left(A_{1}\right)\sim \left(\frac{{\left|a\right|}^{2}+{\left|b\right|}^{2}}{4}+\frac{\left|a\right|\left|b\right|}{2}\cos\left(\varphi_{a-b}\right)\right)\sin^{2}2\theta ,$$
(9)$$I\left(A_{1}\right)\sim I_{⫽}{(A}_{1})+I_{⊥}\left(A_{1}\right),$$
(10)$$I_{⫽}{(E}_{1})∼{\left|c\right|}^{2}\cos\left(\varphi_{a-b}\right)\sin^{2}\left(2\theta \right),$$
(11)$$I_{⊥}{(E}_{1})∼{\left|c\right|}^{2}\cos\left(\varphi_{a-b}\right)\cos^{2}\left(2\theta \right),$$
(12)$$I\left(E_{1}\right)\sim I_{⫽}{(E}_{1})+I_{⊥}\left(E_{1}\right),$$
(13)$$I_{⫽}{(E}_{2})∼{\left|d\right|}^{2}\cos^{4}\theta ,$$
(14)$$I_{⊥}{(E}_{2})∼{\left|d\right|}^{2}\sin^{2}{\theta \cos}^{2}\theta ,$$
(15)$$I\left(E_{2}\right)\sim I_{⫽}{(E}_{2})+I_{⊥}\left(E_{2}\right).$$

From the above formalism, we can know that the phase difference φ_a−b_ between the Raman tensor elements a and b possesses an influence on the Raman scattering intensity of A_1_ mode. From Equation (7), the values of |a|, |b| and phase difference φ_a−b_ in the scattered light intensity I_⫽_(A_1_) are certain. They can be determined through the values of I_⫽_(A_1_) at θ = 0° and 90°, respectively; however, the values of |a|, |b| and phase difference φ_a−b_ in the scattered light intensity I_⊥_(A_1_) are different from that of I_⫽_(A_1_). As known from Equation (8), the values of |a|, |b| and phase difference φ_a−b_ in the scattered light intensity I_⫽_(A_1_) have also influenced the scattered light intensity I_⊥_(A_1_).

Based on the above formalism, we can perform the theoretical calculation and experimental measurements on the 4H-SiC anisotropy Raman scattering. Based on the dynamic atomic theory, different elements of the vibration modes could display variations related to their space structures. In the following, we discuss the anisotropy characteristics and the effects on the Raman tensor of various vibration modes.

### 3.4. Rotation Raman Spectral Intensities and Analyses

Figure 4 exhibits the experimental Raman peak values versus the rotation angle in symbols, and the theoretical curves in solid lines, calculated by Equations (7)–(15), which clearly reveal the interaction relationship of Raman intensity and rotation angle. As shown in Figure 4a, under the case of parallel polarization [y(xx)y-y(zz)y], with the rotation angle increased from 0° to 45°, the Raman intensity of A_1_ mode decreases continually; while with the rotation angle increased from 45° to 90°, the Raman intensity of A_1_ mode increases continually. At the rotation angles of 0°, 90°, 180°, 270°, and 360°, the A_1_ mode has the maxima intensity; while at 45°, 135°, 225°, and 315°, the A_1_ mode disappears. Under the case of perpendicular polarization [y(xz)y-y(zx)y] in Figure 4b, with the rotation angle increases from 0° to 45°, the Raman intensity of A_1_ mode increases continually; while with the rotation angle increases from 45° to 90°, the Raman intensity of A_1_ mode decreases continually. At the rotation angles of 0°, 90°, 180°, 270°, and 360°, the A_1_ mode disappears; while at 45°, 135°, 225°, and 315°, the A_1_ mode has the maxima intensity. These indicate the sinus-like variation of the A_1_ Raman mode intensity depending on the rotation angle and the strong anisotropic characteristics of the 4H-SiC crystal. In addition, to determine the Raman tensor for the A_1_ mode, it can employ the Raman selection rules to perform the theoretical modeling. Both the A_1_ and E_1_ phonon modes are both Raman and IR active, as shown in Figure 4c–d, with similar intensity variation trends for both A_1_ and E_1_ modes; however, the E_2_ mode, with only Raman active, is different from A_1_ and E_1_ modes.

As shown in Figure 4e–f for the strongest E_2_ mode, under the case of parallel polarization [y(xx)y-y(zz)y], with the rotation angle increased from 0° to 90°, the Raman intensity of E_2_ mode increases continually; while with the rotation angle increased from 90° to 180°, the Raman intensity of E_2_ mode decreases continually. At the rotation angles of 0°, 90°, 180°, 270°, and 360°, the A_1_ mode has the maxima intensity, under the parallel polarization; while at near 45°, 140°, 225°, and 310°, the A_1_ mode is weakest. Under the case of perpendicular polarization [y(xz)y-y(zx)y], the Raman intensity variation of E_2_ mode has the same variation trend as the A_1_ mode, while the E_1_ mode possesses a different variation trend from A_1_ and E_2_.

### 3.5. Raman Tensor Element Analyses

Raman intensities of A_1_, E_1,_ and E_2_ phonon modes under the polarization states of parallel [y(xx)y-y(zz)y] and perpendicular [y(xz)y-y(zx)y] are shown in Figure 4. The fitting results for the relationship of Raman intensity and ratio angle under two polarization cases are expressed in Figure 4a–f, with small errors. To determine the Raman tensor elements, the fitting procedure by the least square method and Formulas (7)–(15) with angle variation of function fitting is applied. The best-fitting curves are basically coincident with the experimental data, as shown in Figure 4. The relative values of Raman tensor elements for 4H-SiC crystal are calculated by way of the ratio calculation of fitting parameters are overviewed and shown in Table 1. As applying function fitting with Formulas (7)–(15) on data of Figure 4a–f plus elaboration, proper error bars were obtained and put into Table 1 also. It is found that different Raman modes are completely different. The phase difference and anisotropic ratio of each Raman mode characterize similar information, but there exist phase shifts among different elements. In addition, the values of phase shift and anisotropic ratio are determined from the corresponding Raman mode. Each element of Raman tensor represents its single directional vibration and corelated with another element; therefore, from experimental and theoretical results, the anisotropies of the 4H-SiC sample can be determined.

The 4H-SiC material belonging to a wide bandgap semiconductor compound possesses the wurtzite structure with the hexagonal space group o C^4^_6v_ (P6_3_mc). Based upon the atomic dynamics, by way of Raman scattering measurements, different vibration elements can display the sensitivities correlated to their own space group [42]. The A_1_, E_1_, and E_2_ modes in wurtzite 4H-SiC are Raman active. Among these modes, the A_1_ polarization direction coincides with the *Y*-axis. In a doped semiconductor, the A_1_ mode, by way of the resonant coupling between polarized phonon and plasma (free carriers), exerts an influence on the electronic transport properties [43]. As expressed in Table 1, the relative values of Raman tensor elements for the 4H-SiC crystal are calculated with the help of the ratio calculation on fitting parameters. From Raman scattering spectra under the parallel and perpendicular polarization, the phase difference φ_a−b_ of Raman tensor elements |a| and |b| (i.e., the parallel and perpendicular vectors) for the 4H-SiC crystal are determined as 117.37° and 92.86°, respectively.

It can be seen from the article of Strach et al. [44] that the selection of parameters a, b and phase difference φ_a−b_ is not unique for the perpendicular vectors of the incident and scattered light. So, the ratio of perpendicular polarized vectors in Table 1 is not well defined; however, this ratio is comparatively coincident with the result estimated from the incident and scattered polarized vectors in parallel alignment. For wurtzite material, re-collecting measurements on Raman spectra can confirm the relationship between the anisotropy ratio and the phase difference of Raman tensor elements in wurtzite crystal. W. Zheng et al. [39,40,41] have studied the elucidation of “phase difference” in Raman tensor formalism on wurtzite compounds, namely, AlN, GaN, ZnO, and SiC, and demonstrated the values of phase difference being unanimously confined to around π/2π/2, while the anisotropic ratios display obvious differences in the four compounds. It is reasonable to believe that the phase difference should not be treated as an intrinsic property such as the anisotropic ratio. In addition, it is noted that the anisotropy properties of 4H-SiC materials can also be investigated by way of nanoindentation and scratch experiments [45].

## 4. Conclusions

In summary, from this work for wurtzite 4H-SiC single crystalline material, the rotation, i.e., the angle-dependent Raman scattering (ADSS), has been studied by the combination of experiments and theories. A series of promising and significant results are achieved. The isotropy characteristics were confirmed from the c-face 4H-SiC, while the anisotropy characteristic properties were revealed from the a-face 4H-SiC. The variation functional relationship of the Raman phonon modes versus the angle between the incident laser polarization direction and the parallel (perpendicular) polarization direction was obtained. By way of the parametrization on incident light and scattered light polarization vectors, the selection rules of wurtzite SiC are calculated and well established. Corresponding Raman selection rules are derived according to measured scattering geometries to illustrate the angle dependence. Based upon the selection rules, the intensity variations of the A_1_, E_2,_ and E_1_ modes dependent on the rotation angle are calculated, and the Raman tensor elements of various modes are well-deduced. These detailed theoretical calculation results on the Raman selection role, Raman tensor elements, and the variations of Raman spectral intensities, as well as the phonon anisotropy properties of wurtzite 4H-SiC, are significant and matched well with experimental data.

Our penetrating investigations are helpful to better understand the mechanisms and find ways to further improve the material design and growth of hexagonal SiC, as well as their wide range of applications.

## Figures and Tables

**Figure 1 materials-15-08751-f001:**
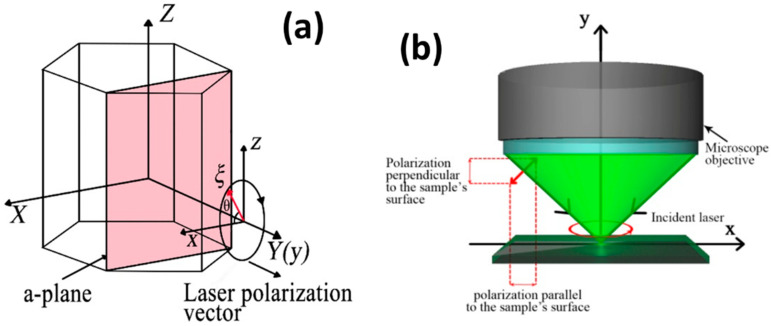
(**a**) The hexagonal wurtzite crystal structure and (**b**) the experimental arrangement. The *Z*-axis is along the c-axis of 4H-SiC crystal, i.e., (0001) direction. The a-plane is shown. The array **ξ** represents the incident light polarizing vector. The θ is the rotation angle with respect to the *X*-axis.

**Figure 2 materials-15-08751-f002:**
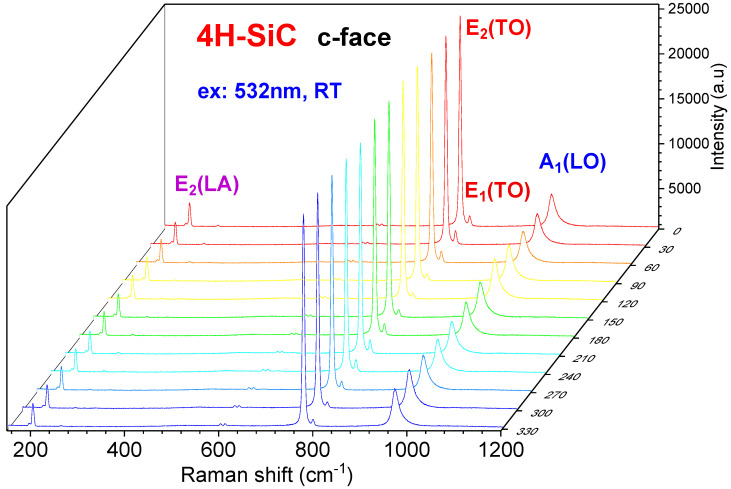
The angle-dependent Raman spectra of the c-face 4H-SiC, at different rotation angles of 0°–330°, were measured at RT with the excitation of 532 nm.

**Figure 3 materials-15-08751-f003:**
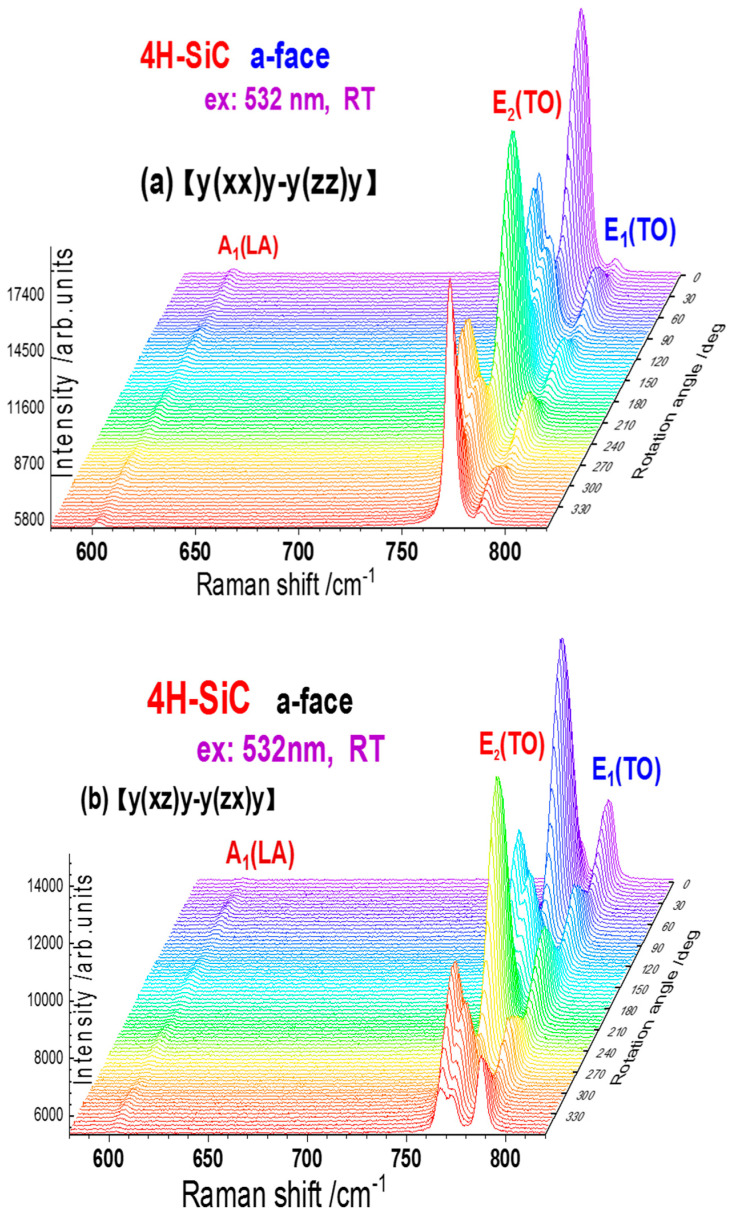
The rotation Raman spectra at the a-plane of 4H-SiC, with the angle varied from 0° to 360° (step size of 5°), under (**a**) parallel and (**b**) perpendicular polarization, respectively.

**Figure 4 materials-15-08751-f004:**
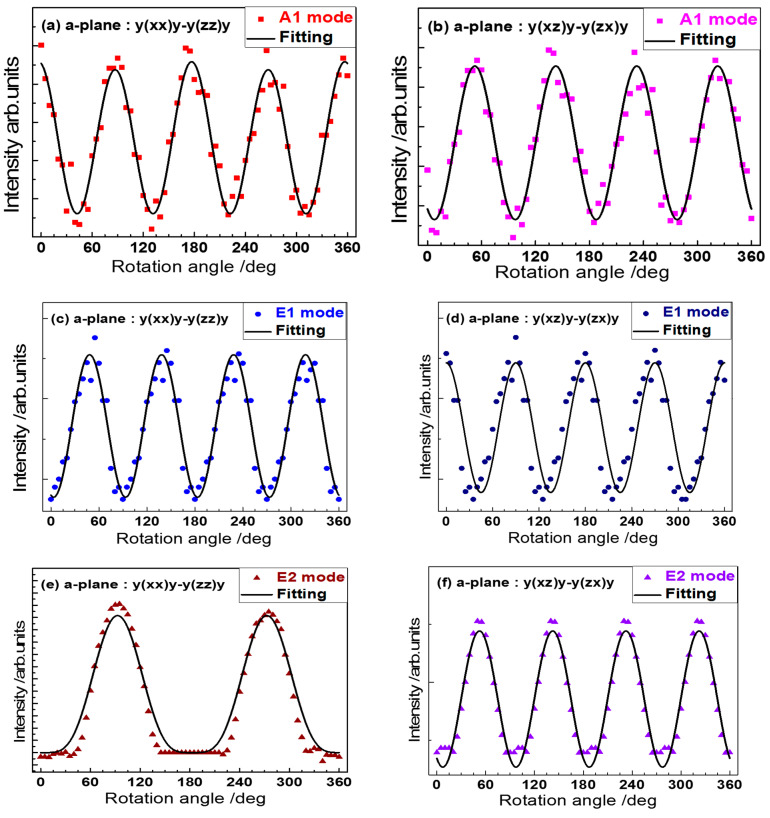
(**a**,**b**) The angle-dependent intensity of the A_1_ signal from the a-plane surface of the SiC sample for parallel and perpendicular polarization vectors of incoming laser and the scattered light. (**c**,**d**) The intensity of the E_1_ signal from the a-plane surface versus the rotation angle for parallel and perpendicular polarization vectors, respectively. (**e**,**f**) The intensity of the E_2_ mode signal versus the rotation angle for parallel and perpendicular polarizations.

**Table 1 materials-15-08751-t001:** The relative values of Raman tensor elements for A_1_, E_1,_ and E_2_ modes of wurtzite 4H-SiC crystal, which are theoretical fitting results for the incident laser (*e_i_*) and scattered light (*e_s_*) under parallel and perpendicular polarization cases, respectively.

A-Plane Tensor Element		$\vec{e_{i}}∥\vec{e_{s}}$ $y(\mathrm{xx})\bar{y}↔y\left(zz\right)\bar{y}$	$\vec{e_{i}}⊥\vec{e_{s}}$ $y(\mathrm{xz})\bar{y}↔y\left(zx\right)\bar{y}$
A_1_ mode	|a/d|	1.003 ± 0.024	0.198 ± 0.019
A_1_ mode	|b/d|	1.004 ± 0.023	0.198 ± 0.020
	$\varphi_{a-b}$	111.37°	92.86°
E_1_ mode	|c/d|	0.388 ± 0.029	0.292 ± 0.013
E_2_ mode	|d/d|	1.000 ± 0.007	1.000 ± 0.006

## Data Availability

The data that support the findings of this study are available from the corresponding author: Z.C.F. (Zhe Chuan Feng), upon reasonable request.

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
