# Peer review of "Angle-Dependent Raman Scattering Studies on Anisotropic Properties of Crystalline Hexagonal 4H-SiC"

_materials, 2022, doi:10.3390/ma15248751_

Round 1
Reviewer 2 Report
In this manuscript, the authors present an interesting characterization process using angle dependent Raman scattering (ADRS) to characterize c-face and a-face 4H-SiC. I have a few concerns regarding the study which I have mentioned as follows;
Minor: Lines 150, 168-170, font needs to be corrected. Actually in several instances. I urge the authors to go over the manuscript thoroughly.
Major: Discussion section, the authors need to include potential application of their work and discuss how this innovativeness fares compared to the previous literature.
It needs more work in terms of discussion section, as to how their work has significantly advanced the knowledge through this project.
Reviewer 3 Report
materials-2041072
Angle dependent Raman scattering studies on anisotropic properties of crystalline hexagonal 4H-SiC
This study conducts angular-dependent Raman Spectroscopy on a-face and c-face 4H-SiC. Combine with theory, 4H-SiC Raman tensor elements were calculated. Overall, this work is interesting and technically sound. The readers should benefit more if the following questions can be addressed.
Question:
1. [table 4] Could authors elaborate more about how the error bars were calculated. It can be easily understood if a case is demonstrated in a supplementary section.
2. [line260] The phase of a and b are determined to be 117.37 and 92.86. Are these phases determined by fitting or materials intrinsic properties? A recent study suggests a phase difference of pi/2 for 6H-SiC, can the same result apply in the case of 4H-SiC? (Photon. Res. 6, 709-712 (2018))
3. Fig. 4(e-f) should discuss about the angle-dependent Raman intensity of E2 mode, I think there are some typo, could authors correct them? [line-217]A1->E2, [line-218] A1->E2, [line-220]E2->E2
4. [line-130, fig. 3] The modes’ peak position in text is slightly shifted comparing with that in figures, e.g. E1(TO) mode at 796.88cm-1, in both fig. 3(a) and (b), E1(TO) is closer around ~788 cm-1? It would be clear if author can provide a figure with only one Raman spectrum indicating all the modes’ positions. This figure can be included in the supplementary.
5. Could author provide the numerical aperture of the 50x objective?
Round 2
Reviewer 1 Report
The authors revised the manuscript according to reviewers comment and the manuscript can be accepted in the current form.
Author Response
Thank you!
Reviewer 2 Report
Accept author's modification in the revised manuscript
Author Response
Thank you.
Reviewer 3 Report
As shown in Figs. 4 (e)-(f) for the E2 mode, under the case of parallel polarization【 229 y(xx)y-y(zz)y】, with the rotation angle increased from 0o to 90o , the Raman intensity of 230 E2 mode increases continually; while with the rotation angle increased from 90o to 180o , 231 the Raman intensity of A1 mode decreases continually. At the rotation angles of 0o , 180o 232 and 360o , the A1 mode has the minima intensity; while at 90o and 270o , the A1 mode is 233 strongest. Under the case of perpendicular polarization【y(xz)y-y(zx)y】, the Raman 234 intensity variation of E1 mode has the same variation trend as the A1 mode
